# Programmable photoacoustic patterning of microparticles in air

Ruoqin Zhang[1,2,5], Xichuan Zhao[3,5], Jinzhi Li[2,5], Di Zhou [2], Honglian Guo [3] ✉,
Zhi-yuan Li [1,4] ✉ & Feng Li [2] ✉

Optical and acoustic tweezers, despite operating on different physical principles, offer non-contact manipulation of microscopic and mesoscopic objects, making them essential in fields like cell biology, medicine, and nanotechnology. The advantages and limitations of optical and acoustic manipulation complement each other, particularly in terms of trapping size, force intensity, and flexibility. We use photoacoustic effects to generate localized Lamb wave fields capable of mapping arbitrary laser pattern shapes. By using localized Lamb waves to vibrate the surface of the multilayer membrane, we can pattern tens of thousands of microscopic particles into the desired pattern simultaneously. Moreover, by quickly and successively adjusting the laser shape, microparticles flow dynamically along the corresponding elastic wave fields, creating a frame-by-frame animation. Our approach merges the programmable adaptability of optical tweezers with the potent manipulation capabilities of acoustic waves, paving the way for wave-based manipulation techniques, such as microparticle assembly, biological synthesis, and microsystems.

The precise handling of micro/nano objects is essential for a wide range of applications in cellular biology[1-4], microbiology[5,6], chemistry[7,8], and micro/nanotechnology[9,10]. Various manipulation techniques have been developed utilizing different mechanisms, including optical[11-16], magnetic[17,18], optic-electric[19-21], and acoustic methods[22-28]. Within these techniques, optical tweezers stand out for their exceptional precision at the nanoscale and extensive utilization[29,30]. Nonetheless, the use of optical tweezers is restricted to objects with specific optical characteristics and small trapping sizes. In contrast, acoustic tweezers provide greater forces and the ability to manipulate much larger objects. Achieving high throughput and arbitrary shape patterning of microscopic objects using acoustic waves is a challenging task[31,32]. Some processes have been made in the advancement of holographic acoustic manipulation through a multi-emitter phased array, but their applications are still constrained by the necessity of a considerable number of transducers and the intricacy of

hardware control[33]. Recently, spatial ultrasound modulation (SUM) has emerged as a promising alternative to spatial light modulation in optical tweezers by utilizing electrode arrays to generate a micro-bubble array in liquid[34,35]. However, the precision of acoustic manipulation based on SUM is limited by the size of the microbubbles.

While integrating optical and acoustic trapping methods can offer more advantages than using them individually[36-40], the fixed trapping positions limit the flexibility of this approach. In the pursuit of flexible and precise manipulations, photoacoustic effect-based tweezers have been proposed[41,42]. This type of tweezers utilizes optically generated acoustic radiation to drive microparticles. However, the formation of random bubbles in the liquid during the process limits the manipulation precision. Although recent studies have explored the use of optically excited Lamb waves to induce the rotation of a single gold foil in nonliquid environments[43], this method is currently restricted to the rotation movement on an optical fiber. Despite advancements in

[1]School of Physics and Optoelectronics, South China University of Technology, 510640 Guangzhou, China. [2]School of Physics, Beijing Institute of Technology, 100081 Beijing, China. [3]College of Science, Minzu University of China, 100081 Beijing, China. [4]State Key Laboratory of Luminescent Materials and Devices, South China University of Technology, 510640 Guangzhou, China. [5]These authors contributed equally: Ruoqin Zhang, Xichuan Zhao, Jinzhi Li.
✉e-mail: hlguo@muc.edu.cn; phzyli@scut.edu.cn; phlifeng@bit.edu.cn

optical and acoustic trapping technology, effectively manipulating a large number of objects independently and simultaneously by combining these methods still presents a significant challenge.

## Results

In this work, we propose a method called programmable photoacoustic patterning (PPAP) for microparticle manipulations. Our system combines the flexibility of optical tweezers with the strength of acoustic waves by utilizing the photoacoustic effect and Lamb wave on a chip-scale platform. To achieve this, we transfer the target patterns to a digital micromirror device (DMD), which modulates the pulsed laser illuminated on it. The modulated laser, carrying the predetermined patterns, is projected onto a multilayer membrane. The membrane absorbs the laser light, leading to rapid thermal expansion. As a result, an elastic wave is generated on the multilayer membrane. The vibration induced by the photoacoustic effect excites the localized Lamb wave on the multilayer membrane. When particles are placed on the membrane, the Lamb wave transfers kinetic energy from the membrane to the particles. The influence of laser power density on the temperature and deformation of the multilayer film is linear. Increasing the power density results in larger displacements of the multilayer membrane, which in turn affects the forces acting on the particles and their motion state. In the air, particles with sufficient kinetic energy can counteract the adhesion between the membrane and particles themselves and, thereby, are propelled toward their designated position. All the physical processes mentioned above occur within 10 ms after the laser pulse excitation. Since the pulse interval of the laser is 100 ms, each pulse can be considered independent (detailed in Supplementary Note 2). The mechanism of PPAP confers a high level of tolerance to fluctuations in particle size and material composition. It can effectively manipulate various particles, such as silica particles ranging from 25 to 55 μm in diameter, zirconia particles from 30 to 100 μm, pollen particles with a diameter of 50 μm, and so on (detailed in Supplementary Note 3).

Our system, as depicted in Fig. 1a and described in detail in the "Methods" section, utilizes a 6 ns 532 nm pulse laser with design patterns that are modulated by DMD. The laser is projected onto the interface between water and a TiN−steel−graphite multilayer membrane. The black TiN film in the membrane absorbs the laser light, causing rapid thermal expansion, which generates an elastic wave on the multilayer. The water, as shown in Fig. 1a is used for damping of the elastic waves, leading to a localized wave. The vibration caused by the photoacoustic effect excites the antisymmetric Lamb wave on the multilayer, which appears as undulations spreading across the surface of the layer. The vibrations weaken by three orders of magnitude after 10 μs, indicating that the vibration is localized around the area of excitation light. When a pulsed laser is directed at a rectangular area 0.4 mm in the $x$ direction and 4 mm in the $y$ direction on the multilayer membrane, it generates an elastic wave on the membrane. Subsequently, this wave propagates outwards in both positive and negative $x$ directions. Figure 1b and c illustrate the simulation and experimental measurements of the membrane's deformation in the $z$ direction upon the illumination of the pulsed laser at a power density of 1 MW/cm². The horizontal axis in Fig. 1b and c represents the position from the center of the illuminated region, while the vertical axis is the time of elastic wave propagation. It can be concluded that elastic waves generated by optoacoustic effects propagate through space at an approximate speed of 60 m/s. The measured result, as shown in Fig. 1c is in good agreement with the simulated one in Fig. 1b. Through theoretical derivation; we can obtain the space−time diagrams of the

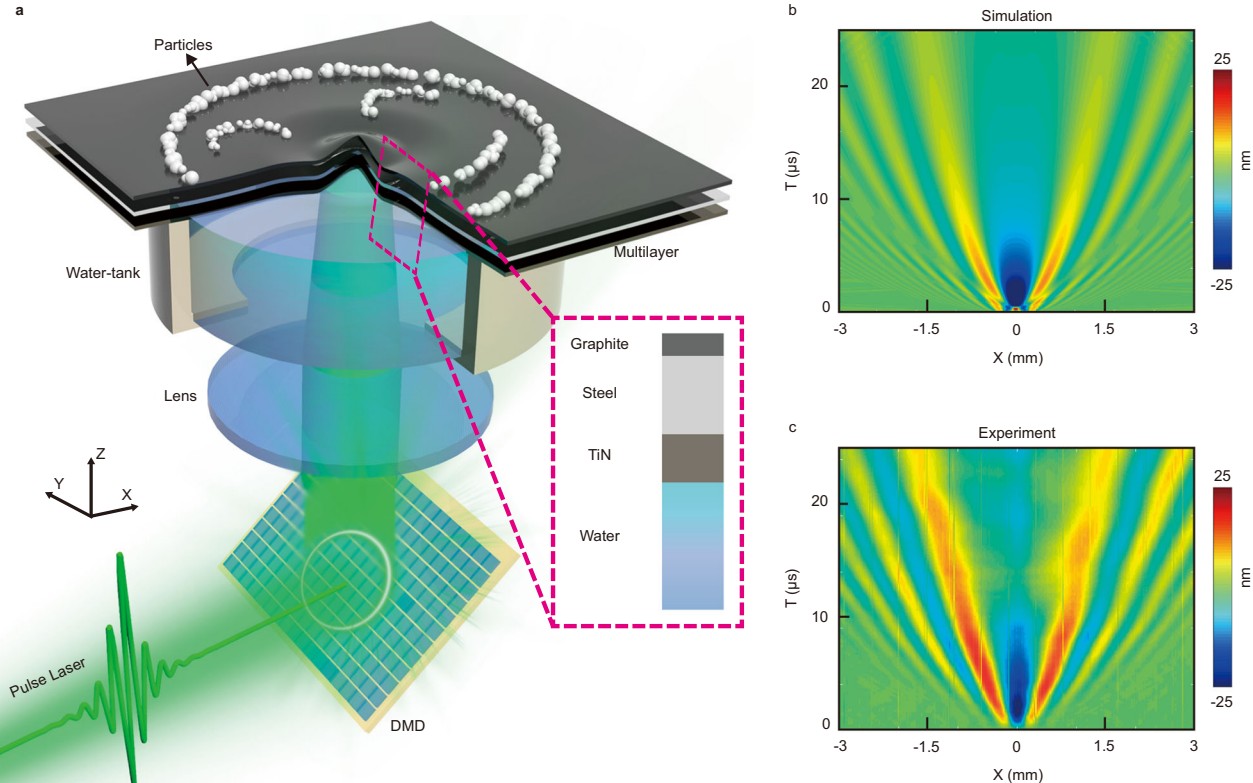

**Fig. 1 | Schematic diagram of programmable photoacoustic patterning setup and the propagation of the elastic waves. a** The setup showing the pulsed laser beam modulated by the DMD and irradiating the TiN−steel−graphite membrane to induce a local photoacoustic effect. **b** Simulation results depicting space−time diagrams of elastic waves induced by the photoacoustic effect. **c** Experimental measurements of elastic wave propagation depicted in space−time diagrams.

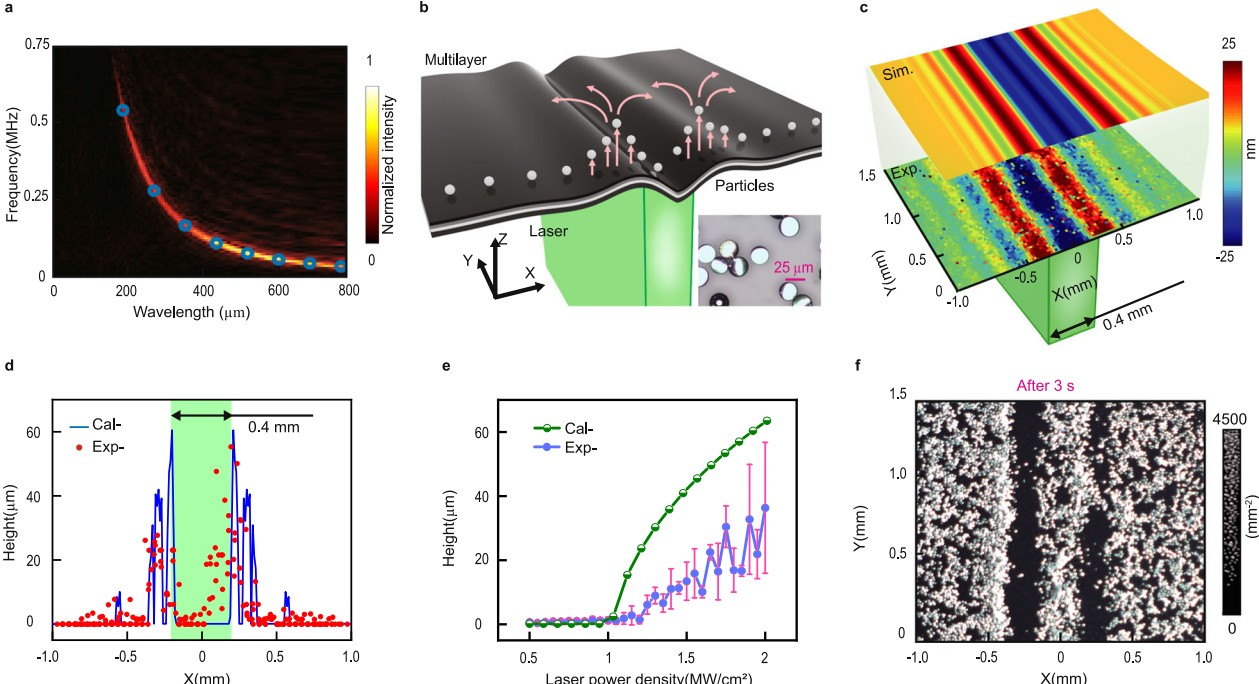

**Fig. 2 | The movement of particles driven by opto-acoustically excited elastic waves. a** Dispersion relationship between the frequency and wavelength of the elastic waves, represented by the color map (measured) and blue circles (calculated). **b** Schematic representation of particle jumping induced by photoacoustic effect. The inset is micrographs of 25 μm silica particles. **c** Simulated and experimental results showing the deformation of the membrane caused by laser radiation within a rectangular area of 0.4 mm in width and 4 mm in length. **d** Relationship between maximum jumping heights and horizontal position of particles. calculation results are represented by a blue curve; experiment results are represented by red dots. The light green rectangle area represents the laser-illuminated region. **e** Relationship between particles' jumping height and laser energy of each pulse. The green semicircles represent the calculation results, while the blue dots represent the average jump height and error bars are plotted by purple line segments. **f** Particle patterning 3 s after initial laser irradiation.

membrane's deformation similar to Fig. 1b, c. The results are shown in Supplementary Fig. 3.

Then, we simulated the space–time evolution of vibrations on a multilayer membrane under different widths of laser stripe excitation. Combined with the experimental results, we found that increasing the width of the laser stripe results in a decrease in pattern resolution and an increase in vibration strength, as shown in supplementary Note 5. Additionally, when the width of the laser stripe exceeds 430 μm, it results in vibration overlap, while a width below 280 μm leads to insufficient vibration excitation, ultimately affecting the clarity of the patterns. The width of 400 μm is proven to be an optimal value and corresponds to a pattern resolution of 270 μm.

To assess the precision of PPAP and characterize the elastic waves in the multilayer membrane, we conducted two-dimensional Fourier transforms on the space–time elastic wave fields depicted in Fig. 1c. The color map of the Fourier transform diagram depicted in Fig. 2a, while the dispersion diagrams are represented by the high values in the color map. These high values indicate the dispersion relationship between frequencies and wavelengths of elastic waves. The blue circles in Fig. 2a depict the analytical dispersion of Lamb waves calculated using Eq. (1) in the "Methods" section. Our analysis reveals that the photo-acoustically excited elastic waves in our PPAP system are Lamb waves, with wavelengths ranging from approximately 0.2–0.8 mm and frequencies ranging from about 0.1–1.0 MHz, as confirmed by the color map in Fig. 2a, which exhibits good agreement with the analytical dispersion of Lamb waves.

The pulsed laser beam, modulated by DMD, excites Lamb waves in the multilayer, with their exaggerated deformation shown in Fig. 2b. To showcase the ability of PPAP, we use silica particles with an average diameter of 25 μm, as shown in the micrographs (in the inset of Fig. 2b). The vibration of a multilayer membrane involves the conversion of

kinetic energy and potential energy. When the particles are placed on the multilayer, the membrane that deforms upward will transfer energy to the particles in this deformed region. If the kinetic energy of the particle is sufficient, it will detach from the multilayer membrane and fly into the air. In Supplementary Note 4, we analyzed the different force states of particles on the multilayer film and the characteristics of the detachment critical point. Then, using numerical integration methods, we obtained the variation of the force exerted on the particles with time and their motion state. Particles that meet the separation critical condition are lifted, then scatter, and fall onto the surrounding areas. This process creates empty spaces in the layer of particles, causing falling particles to accumulate around these voids.

The deformation map, as shown in Fig. 2c from the simulation, indicates that the highest deformation occurs at both edges of the irradiated rectangular region with a width of 0.4 mm. In Fig. 2c, the measured deformation distribution is presented below the simulation. The results illustrate that 2.0 μs after the laser pulse illumination; the center region is visibly dented, while the edge region is protruding. The main force that hinders the separation of particles from the membrane is the van der Waals force, which is tens of thousands of times stronger than the gravitational force acting on the particles. Based on Supplementary Notes 4 and 5, we can calculate the spatial distribution of the maximum jumping height of particles. The calculated results with a laser power density of 2 MW/cm² are depicted by the blue curve in Fig. 2d, demonstrating the agreement with the experimental results represented by red dots. The jumping heights of particles vary depending on their positions relative to the laser-illuminated region. The sparse distribution of particles decreases the interactions between them, facilitating their return to their original positions after jumping. If a thick layer of particles is spread over the multilayer membrane, the particles will collide and scatter in various

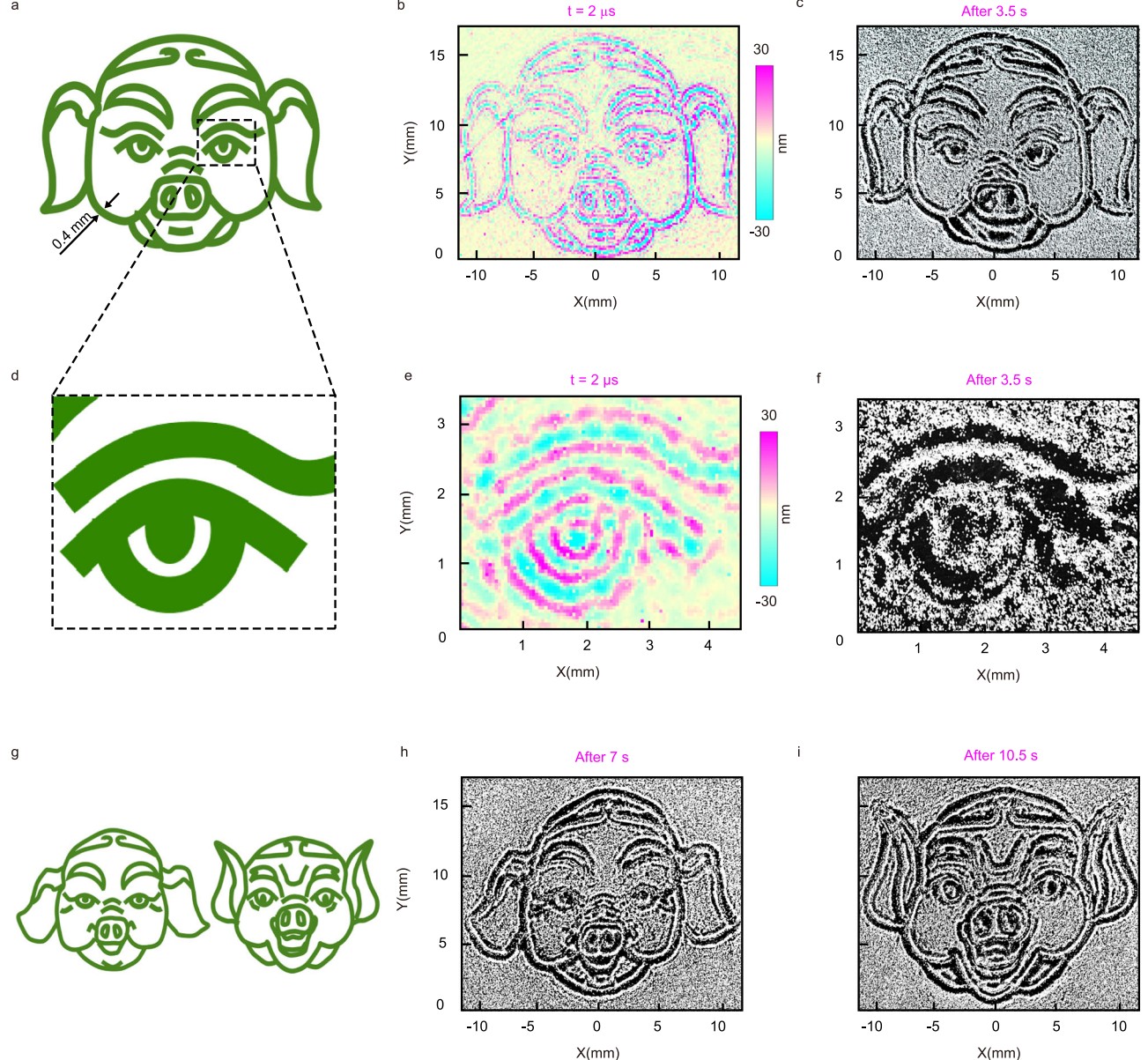

**Fig. 3 | Intaglio style patterning of Pigsies. a** The target image of Pigsy.
**b** Measured deformation fields of the membrane in the *z* direction obtained after
2 μs of the laser pulse illumination. **c** Picture of the patterned particles corre-
sponding to the first target Pigsy. **d** Zoomed-in figure of first Pigsy' right eye.
**e** Deformation of Lamb wave fields of Pigsy's right eye. **f** Zoomed-in patterning
picture corresponding to Pigsy's right eye. **g** Second and third target Pigsy pattern
and laser-illuminated region. **h, i** Particle patterning pictures of the left and right
target Pigsy patterns in (**g**), respectively.

directions. As a result, the particles within the irradiation area will be
rearranged a few seconds after the initial laser pulses.

The laser power linearly determines the strength of vibration,
which, in turn, influences the jumping height of particles and the final
arrangement of particles. Based on the information about vibrations
from the simulation results using different laser power densities, we
calculated the maximum jumping height of particles using numerical
integration methods. The results, represented by the green semicircles
in Fig. 2e, exhibit slight differences compared to the experimental
results represented by the blue dots. This discrepancy may be attrib-
uted to the omission of the camera's measurement error and particle
rotation during the jumping process in the calculations. Both experi-
mental and calculated results demonstrate that the minimum laser
power density required for particles to jump is 1.2 MW/cm². 

Figure 2f is the patterned particles with a density of ~4500 mm⁻²
when the pulsed laser illuminates on a rectangular area of 0.4 mm ×
4 mm. These patterned dark-white-dark band features allow for the
creation of complex designs, which are utilized to generate intricate
patterns in subsequent steps. By considering the mass of the particles
$2 \times 10^{-8}$ g and the maximum acceleration of the particles $4 \times 10^{-6}$ m/s², 
we estimate that the pushing force generated by PPAP is around 80 μN,
which is comparable to the force provided by traditional acoustic
tweezers. However, the trapped particles by PPAP are much smaller in
size (~25 μm) compared with those trapped by the acoustic tweezers
in air[44].

We utilize PPAP to create complex particle patterns inspired by
the art of sand painting. Unlike traditional point-by-point painting
methods, our approach allows the entire pattern to be developed
quickly, transitioning from blurry to clear within several seconds. To
illustrate the capabilities of this approach, we created three static Pigsy
figures with varying expressions, ranging from calm to joy. Figure 3a
displays the first target image that is used as the pattern for PPAP. The

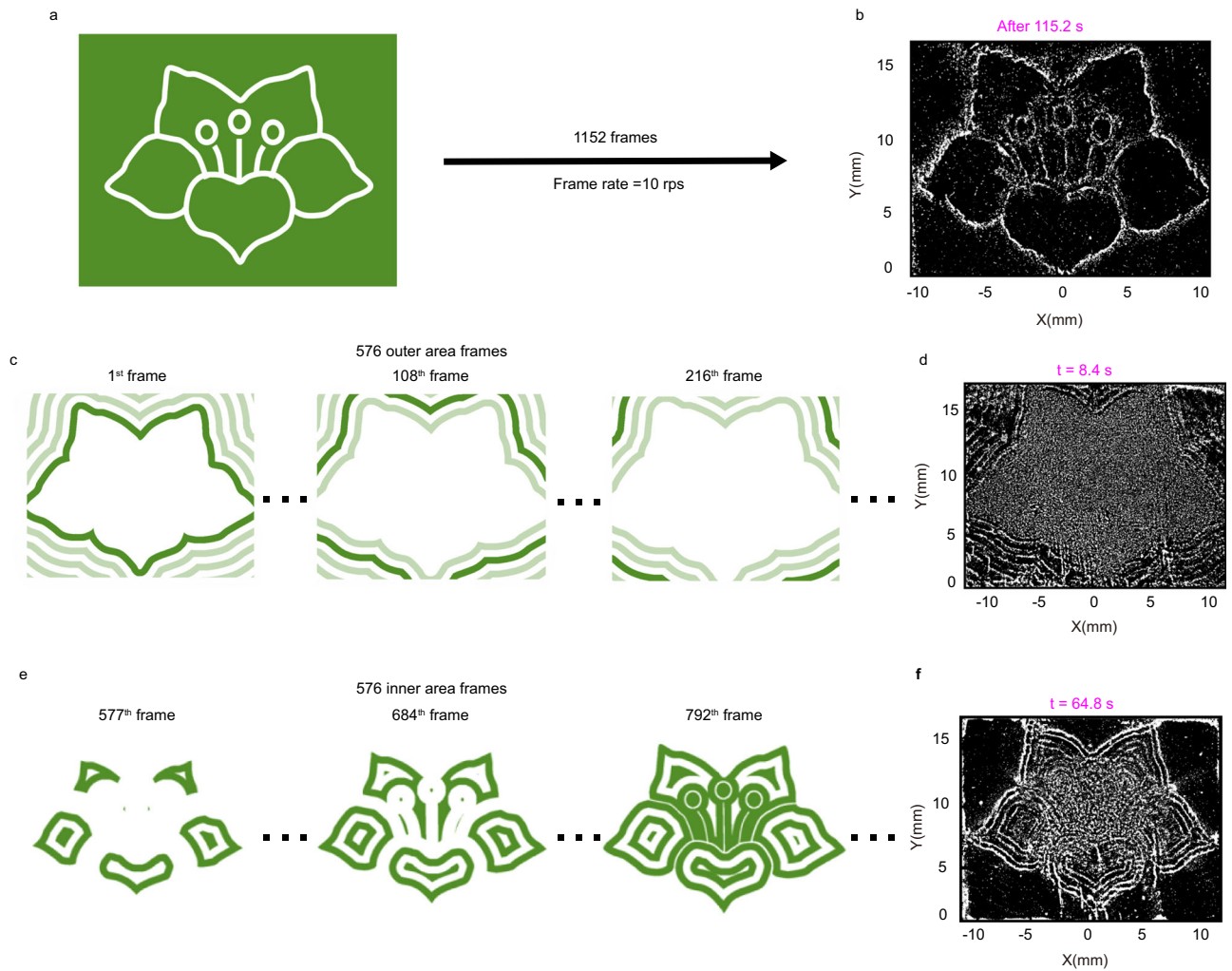

**Fig. 4 | Letterpress style kapok flower patterning. a** The letterpress style target image. **b** The final results of the letterpress style flower patterning. **c** Three sequential frames of the outer area pattern. We blurred certain lines to highlight the evolving trajectory of the innermost lines of the first frame. **d** The process of removing particles from the outer area of the flower. **e** Three sequential frames of the inner area pattern. **f** The progress of driving particles in the inner area of the flower.

width of the lines drawn is set to 0.4 mm. Figure 3b shows the deformation distribution of the membrane obtained after 2.0 µs of the laser pulse illumination, which maps the first target Pigsy image. As seen in Fig. 3b, each curve of the target image causes dents and bulges on the multilayer, similar to the behavior observed in Fig. 2c. The elastic waves are confined to the vicinity of the laser irradiation region, resulting in patterns that duplicate the target image. Before initiating the PPAP process to create a pigsy pattern, the membrane is uniformly coated with particle powder with a number density of ~4500 mm$^{-2}$. After 3.5 s, the dark-white-dark curves form a clear Pigsy intaglio style pattern in a 22 × 16 mm$^2$ area, as shown in Fig. 3c and the formation process shown in Supplementary Movie 1. This demonstrates the ability of our PPAP system to arrange particles into complex and precise patterns over a large area

Furthermore, we demonstrate that PPAP enables us to clearly capture the details of the image through pattern generation. Figure 3d shows the zoomed-in right eye of the Pigsy image. Figure 3e is the local deformation fields of the eye, with a line resolution of about 270 µm. The intaglio style pattern of the local right eye part corresponding to Fig. 3c is displayed in Fig. 3f. Additionally, we demonstrate the versatility of PPAP by arranging pine pollen into the Pigsy pattern (see Supplementary Note 8). Figure 3g shows the second and third target images, while Fig. 3h and i are the pictures of the patterned particles corresponding to the second and third image frames, respectively. As anticipated, the image undergoes a dynamic change process, showcasing the ability of PPAP to extend painting to movies.

Intaglio-style patterning involves only removing particles from the designated positions, while letterpress-style patterning requires additional steps. In letterpress-style patterning, we need to remove particles from the background while keeping particles at the target positions. As an example, to produce the kapok flower sketch depicted in Fig. 4a, we need to eliminate particles located in the green area while preserving the particles that form the flower outline, as indicated by the white curves. The particles tend to gather near the center of the stripe. Therefore, we can manipulate the particles on the multilayer film by moving the laser stripe (for detailed information, refer to Supplementary Note 6).

To achieve the letterpress style pattern of a kapok flower, as shown in Fig. 4b, dynamic playback of 1152 frames of images at a frame rate of 10 fps is required using the DMD. The laser pulses emitted by the laser system are synchronized with the pattern sequence. As a result, the laser pattern projected onto the multilayer film changes at a rate of 10 fps. The design process of the image sequence is depicted in

Supplementary Note 7. The first 576 frames are used to remove particles from the external background of the kapok flower pattern, while the remaining 576 frames are used to transport particles from the inner to its contours.

The membrane is initially coated uniformly with the silica particles as above mentioned. Figure 4c illustrates the first 576 frames, where the curves of the outer area sequentially expand outward, and the PPAP removes particles located in the outer area (see Supplementary Movie 2). The picture of the particles driven by the PPAP after 8.4 s of the initial laser pulse is shown in Fig. 4d, where particles outside of the flower are gradually eliminated. After 57.6 s, the particles in the outer region are almost removed.

The remaining 576 frames are shown in Fig. 4e, where the inner region is decomposed into multiple connected domains, each of which expands outward independently. Figure 4f displays the state of the patterning process after 64.8 s of the initial laser pulse. The internal region is partitioned into various interconnected domains by curves, and the particle ripples are clearly visible.

The intaglio style and letterpress style figures presented above demonstrate the versatility of the PPAP method, showcasing its ability to create complex particle patterns over a large area. It is worth noting that PPAP can manipulate not only a large number of particles but also a smaller number of particles. For instance, PPAP can accurately move 30 silica particles along a specified path (see Supplementary Movie 3 and Note 8) to a distance of 10.0 mm with a speed of 0.35 mm/s and an accuracy of ~120 µm. These results highlight the flexibility and efficiency of PPAP, which are comparable to those of holographic optical tweezers, regardless of the particles' optical properties.

## Discussion

In this study, we have introduced PPAP, a technique for efficiently manipulating particles on a multilayer membrane in air based on the photoacoustic effect and Lamb waves. Large numbers of silica particles are patterned into the desired figure on a multilayered membrane. By rapidly and successively modulating the field distribution of the laser, the particles within the area of illumination can dynamically flow along the corresponding elastic wave field, resulting in a frame-by-frame animation effect. PPAP possesses both the flexibility of optical hologram tweezers and the strength of acoustic tweezers. Additionally, increased resolution and quality may be achieved by improving optical systems with low aberration and laser equipment with superior uniformity. Utilizing image recognition self-feedback can further enhance the quality of patterned images. These patterned particles may be transferred layer-by-layer to create a 3D structure through micro 3D printing (detailed in Supplementary Note 10). Furthermore, PPAP techniques can be applied in liquid environments, enabling the controlled manipulation of targets within liquids.

The high-precision pattern/printing technology presented in this study has a broad range of potential applications, including microparticle assembly, micro-robot manufacturing, and organ culture. Our research has opened up possibilities for efficient and high-resolution particle manipulation, and the overall transfer of patterned particles provides a pathway for micro-3D printing. The combination of PPAP with closed-loop control and layer-by-layer transfer has the potential to revolutionize the field of micro/nano manufacturing.

## Methods

### The experiment setup

In our PPAP system, elastic wave fields are generated by employing a pulsed laser beam with a wavelength of 532 nm, which is modulated by a digital micromirror device (DMD). The laser (Nimma-900 laser from Beamtech Optronics Co., Ltd.) has a pulse width of 6 ns, a single pulse energy of about 480 mJ, and a repetition frequency of 10 Hz. The laser beam passes through a convex lens ($f$ = 40 cm) and a 1 mm-

thick layer of water. The laser beam then illuminates a multilayer membrane consisting of a 30 nm-thick black titanium nitride (TiN) layer, a 5 µm-thick stainless-steel membrane, and a 10 nm graphite layer. The graphite layer is used to reduce the adhesion force between the particles and the multilayer membrane. The substrate of the membrane is 304 stainless steel, with a size of 50 mm × 50 mm, and the back side of the steel plate is coated with TiN by electric beam evaporation deposition. The front side of the substrate is coated with graphite by sputtering coating. The particle we used is silica, with a density of 2500 kg m$^{-3}$. The DMD chip (TI S1410-9032) has a size of 0.95 inches and a resolution of 1400 × 1050 pixels. The deformation distribution shown in Figs. 1b, c and 2c are measured by a laser vibrometer (Polytech VFX-F-110). A high-speed camera (Flare 2M360MCL, IO Industry) captures the jumping behavior of the particles, as shown in Fig. 2d, e, while another camera (MV-CE200-10UC, Hikvision) records the patterning process, as shown in Figs. 3c, h, i and 4b, d, f.

### Theoretical derivation of photoacoustic

The photoacoustic effect involves the conversion of light energy into heat energy and heat energy into mechanical energy. In the derivation of the opto-thermal process, we first calculated the temporal variation of the surface temperature of the multilayer membrane. Using this as a boundary condition, we then computed the diffusion of heat and obtained the spatiotemporal distribution of the temperature within the multilayer membrane. By calculating the strain caused by temperature changes, we obtained the initial displacement. The multilayer film under laser stripe illumination can be approximated as an infinitely long beam. We then applied the initial displacement to the vibration equation of the infinitely long beam to obtain the space–time diagrams of the membrane's deformation. From the derivation, We find that the laser power density linearly affects the temperature and vibration displacement of the multilayer membrane (detailed in Supplementary Note 1).

### Photoacoustic simulation

We used the COMSOL Multiphysics solver package to perform simulations of pulsed laser-excited Lamb waves. The simulation involves four modules: radiation beam heat absorption, solid (fluid) heat transfer, solid mechanics, and pressure acoustics, as well as three multi-physics fields: radiation heat transfer in the absorbing medium, thermal expansion, and acoustic-structural boundary. The width of the water layer and membrane in the simulation were both 6 mm, with the thicknesses of the water and membrane being 100 and 5 µm, respectively. To streamline the simulation process, we simplify the multilayer membrane as a single stainless-steel material with the TiN light energy absorption ratio. The density of the layer is 7930 kg m$^{-3}$, Young's modulus is 210 GPa, the Poisson's ratio is 0.3, the specific heat capacity is 0.5 kJ kg$^{-1}$ K$^{-1}$, the heat conductivity is 45.0 W m$^{-1}$ K$^{-1}$ and the optical absorption ratio of the layer is $7 \times 10^7$ m$^{-1}$. The detailed simulation process is shown in Supplementary Note 2.

### The frequency of the anti-symmetric Lamb wave

The Lamb wave mode occurs when the thickness of the vibrating material is much smaller than the wavelength. When there is water on one side of the membrane, the frequency of the anti-symmetric Lamb wave can be obtained by

$$f = \frac{h_s}{2\pi} \sqrt{\frac{E}{12\rho(1-\nu^2)}} \sqrt{\frac{\rho \cdot h_s}{\rho \cdot h_s + \rho_w \cdot \delta_w}} \cdot \left(\frac{2\pi}{\lambda}\right)^2 \qquad (1)$$

where $h_s$ is the thickness of the stainless membrane, and $\rho_w$ is the density of water. $\delta_w$ is the evanescent Lamb wave penetration depth in water. The dispersion of the Lamb wave based on Eq. (1) is represented by circles shown in Fig. 2a.

## Data availability

All data supporting the findings of this study are available within the article and its supplementary files. Any additional requests for information can be directed to and will be fulfilled by the corresponding author.

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

## Acknowledgements

This work is supported by the National Natural Science Foundation of China (Grants No. 12074446, received by H.G., No. 12272040, received by F.L., and No. 12102039 received by D.Z.).

## Author contributions

F.L. conceived and designed the research. R.Z., X.Z., and J.L. carried out the experiments. Z.-Y.L. and D.Z. analyzed the results. R.Z., F.L., and H.G. wrote the manuscript. H.G., Z.-Y.L., and F.L. supervised the project. All the authors contributed to the discussions of the results and the manuscript preparation.

## Competing interests

The authors declare no competing interests.
