## [Peer Review File · Nature Communications]

Programmable Photoacoustic Patterning of Microparticles in AirReviewers' Comments:

Reviewer #1:

Remarks to the Author:

The manuscript presents a programmable approach that combines optical and acoustic contactless manipulation of particles. The use of hybrid acoustic-optical approaches for trapping and manipulation has been presented several times, reducing the novelty and significance of this work, which may not meet the high standards of this journal.

Few examples:

- G. Thalhammer, C. McDougall, M. P. MacDonald, M. Ritsch-Marte. Acoustic force mapping in a hybrid acoustic-optical micromanipulation device supporting high resolution optical imaging <https://doi.org/10.1039/C6LC00182C>

- M. W. Haakestad and H. E. Engan, "Acoustooptic properties of a weakly multimode solid core photonic crystal fiber," *J. Lightwave Technol.*, vol. 24, no. 2, pp. 838–845, 2006, <https://doi.org/10.1109/jlt.2005.862431>

- M. W. Haakestad and H. E. Engan, "Acoustooptic characterization of a birefringent two-mode photonic crystal fiber," *Opt. Express*, vol. 14, no. 16, pp. 7319–7328, 2006, <https://doi.org/10.1364/oe.14.007319>

However, previous hybridisation approaches are only used for fixed trapping and positioning and are not able to dynamically position particles. In particular, the authors have proposed programmable optical-acoustic patterning (POAP) for microparticle manipulation by exploiting the photoacoustic effect and the Lamb wave on a chip-scale platform in an innovative way.

The methodology and results are clearly described, but it is not clear whether there is an inertia that affects the thermal expansion of the membrane and how this may affect the dynamic assembly of the particles. In other words, how quickly can the geometry of the pattern be changed or reversibly returned to an initial configuration? How does this affect the dynamic tuning of the particle assembly? Out of curiosity, how might the weight of the particles affect the assembly kinetics? Are there any mass, size or shape limits for the particles used in this process? the author should discuss these aspects better, which could strengthen the impact of the paper.

Finally, the authors make a claim about possible applications in biological systems. Such a claim needs to be slowed down or even removed as it does not demonstrate feasibility or proof of concept for potential use in the biological field. I would suggest that proof of use in biological systems should be provided by testing the assembly of particles, cells or microtissues in liquid media or other biological environments. Alternatively, the authors need to demonstrate how the assembly of particles in air can be used for biological purposes.

Reviewer #2:

Remarks to the Author:

In this work, the photoacoustic effects have been used to generate localized Lamb wave fields that can mimic arbitrary laser pattern shapes. By using localized Lamb waves to vibrate the surface of the multilayer membrane, authors can pattern tens of thousands of 25µm - diameter silica particles into the desired pattern simultaneously. The results presented in the manuscript are convincing, but the topic is not new, the mechanism explanation is weak, and lack of convincing and exciting application demonstration. Thus, this manuscript in the current version cannot be accepted to be published in nature communications.

Some questions need to be answered:

Since this manuscript is focused on particle control via lamb wave generated by the photoacoustic method, the title "optical-acoustic patterning" is misleading the reader. Optical-acoustic patterning is a kind of means the author wants to pattern the particles via coupling the optical field and acoustic field. Particle manipulation has been proposed by many published papers, compared with these papers, this manuscript lacks applications. As the author said in the abstract, this work can pave the way for new wave-based manipulation techniques, such as microparticle assembly, biological synthesis, and

microsystems, can the author provide at least one of the proposed applications to prove the value of this research?

What is the resolution for the proposed method and how will the author identify it from the current method?

For the mechanism discussion, how did the 25 μ m particle was chosen to run the experiment? Can the author use other particles with different diameters to demonstrate the control mechanism?

Figure 3 and Figure 4 almost deliver a similar message, one does not see the technology, mechanism, or application as different.

The author provides that "the Lamb wave transfers kinetic energy from the membrane to the particles." One believes that should be true, this experiment phenomenon is kind of like the Chladni plate experiment, the author may provide a more accurate and detailed theoretical explanation to support the experimental result.

Reviewer #3:

Remarks to the Author:

Comments: In this manuscript, the author used photoacoustic effects to generate localized Lamb wave fields that can mimic arbitrary laser pattern shapes. By using localized Lamb waves to vibrate the surface of the multilayer membrane, they can pattern tens of thousands of 25 μ m in-diameter silica particles into the desired pattern simultaneously.

The manuscript is well-organized, and this process is good and may attract broad interest in the field of acoustics. However, several issues still need to be addressed before consideration for publication.

1. It is mentioned in the manuscript that POAP technology can realize the dynamic flow of particles by quickly and continuously adjusting the laser shape, thereby creating frame-by-frame animation effects. The authors should explain and analyze the mechanism of this active flow in detail to ensure that the particles can flow accurately along the corresponding elastic wave field.
2. The authors mentioned that POAP technology can efficiently manipulate particles on multi-layer films through the photoacoustic effect and Lamb wave in the air. The authors should conduct a detailed study and analysis of the mechanical properties during this manipulation. For example, how does the laser's power and frequency affect particle manipulation?
3. The authors claimed that POAP technology can pattern many particles into the desired pattern. Did the authors evaluate the precision and reproducibility of this modeling procedure? Are there any errors or deviations? If so, have the authors proposed methods or measures for improvement?
4. It is mentioned in the manuscript that POAP technology can be combined with closed-loop control and layer-by-layer transfer to realize a revolution in the field of micro/nano manufacturing. May I ask if the authors have explored or verified the practical application of this combination? Has it achieved practical application results in the fields of particle assembly, biosynthesis, or microsystems?
5. The authors mentioned that POAP technology can be applied in the liquid environment to realize the control and manipulation of the target in the liquid. May I ask which fluid environment the author chooses for the experiment and whether there is a big difference in the experimental results of different fluid environments?
6. There are many grammatical and writing errors in the manuscript. Please correct them before publication.

Reply to Reviewer #1:

The manuscript presents a programmable approach that combines optical and acoustic contactless manipulation of particles. The use of hybrid acoustic-optical approaches for trapping and manipulation has been presented several times, reducing the novelty and significance of this work, which may not meet the high standards of this journal.

Few examples:

• *G. Thalhammer, C. McDougall, M. P. MacDonald, M. Ritsch-Martel. Acoustic force mapping in a hybrid acoustic-optical micromanipulation device supporting high resolution optical imaging <https://doi.org/10.1039/C6LC00182C>*

• *M. W. Haakestad and H. E. Engan, "Acoustooptic properties of a weakly multimode 2006, <https://doi.org/10.1109/jlt.2005.862431>*

• *M. W. Haakestad and H. E. Engan, "Acoustooptic characterization of a birefringent two-mode photonic crystal fiber," *Opt. Express*, vol. 14, no. 16, pp. 7319–7328, 2006, <https://doi.org/10.1364/oe.14.007319>*

However, previous hybridisation approaches are only used for fixed trapping and positioning and are not able to dynamically position particles. In particular, the authors have proposed programmable optical-acoustic patterning (POAP) for microparticle manipulation by exploiting the photoacoustic effect and the Lamb wave on a chip-scale platform in an innovative way.

We appreciate the reviewer for the comment on our manuscript. The recommended literatures are valuable sources of inspiration for us. Drawing upon these references, we have conducted further research to enhance the overall quality of our work. We have revised the manuscript extensively and provide point-by-point responses to the comments.

1)The methodology and results are clearly described, but it is not clear whether there is an inertia that affects the thermal expansion of the membrane and how this may affect the dynamic assembly of the particles. In other words, how quickly can the geometry of the pattern be changed or reversibly returned to an initial configuration? How does this affect the dynamic tuning of the particle assembly?

We sincerely appreciate the reviewer for bringing up the unclear statements in our manuscript. To address the physics involved after 6 ns irradiation of the pulsed laser on the multilayer membrane, we have performed calculations to determine the transient temperature rise, the displacement of the multilayer membrane, and the particle motions. These results are presented in Fig. R1. The multilayer membrane absorbs the energy from the laser, resulting in a temperature increase. The temperature rise induces thermal expansion, leading to the propagation of Lamb waves in the multilayer membrane. The vibration at the position of $x = 0\text{mm}$ ceases after 10 μs . Subsequently, the multilayer

membrane has an impact on the particles, causing them to be dislodged. The jumping time of the particles is within 10 ms. Since the pulse interval of the laser is 100 ms, each pulse can be considered independent, allowing for the analysis of the physical phenomena under the condition of single pulse excitation.

Hence, it can be concluded that the dynamic assembly process of the particles following each pulse does not interfere with one another. As a result, the dynamic formation of the intaglio pattern, as depicted in Fig. 3c, h, i necessitates several tens of independent pulses. In addition, the formation of the letterpress pattern shown in Fig. 4g requires over a thousand pulses in order to achieve the desired outcome.

Fig. R1. (appears as S. Fig. 1 of the Supplementary Information) Evolutions of laser power density, transient temperature rise, displacement of the multilayer membrane, and particle motions over time.

Summary of the modifications:

(1) On Page 3, marked by deep red.

“All the physical processes mentioned above occur within 10 ms after the laser pulse excitation. Since the pulse interval of the laser is 100 ms, each pulse can be considered independent (detailed in Supplementary Note 1).”

(2) We have added Figs. R1 in the revised Supplementary Information, on Page 3, Fig. S1, to clarify the time scale of photoacoustic manipulation.

2) Out of curiosity, how might the weight of the particles affect the assembly kinetics? Are there any mass, size or shape limits for the particles used in this process? the author should discuss these aspects better, which could strengthen the impact of the paper.

We would like to express our appreciation to the reviewer for posing a pertinent question regarding the influence of particles' properties. In an air environment, the primary force governing the interaction between micron-sized particles and a flat surface is the van der Waals force. It is important to note that the van der Waals force

is tens of thousands of times magnitude of the gravitational one.

The van der Waals force between spherical particle and a surface is as follows:

$$F_{vdw} = \frac{A \cdot r}{6d^2} + \frac{A}{6\pi d^3} \cdot \pi z^2,$$

where A represents the Hamaker constant, r is the radius of particle, d is the separation distance between the particle and the surface, and z is the radius of the contact area between the particle and the surface.

	25 μm silica particle	40 μm silica particle	55 μm silica particle
G	0.14 nN	0.59 nN	1.54 nN
F_{vdw}	46.58 μN	55.95 μN	95.50 μN

Tab. R1. The Comparison of gravity and Van der Waals force of silica particles with diameters of 25 μm , 40 μm , and 55 μm , respectively.

The magnitude of the van der Waals force is intricately linked to the material properties of both the surface and the particles, as well as the sizes of the particle and the contact area between the particle and the surface. In Fig. R2a and b, we present a comparison of the quality of ring patterns formed by different particles under a laser power density of 1.5 MW/cm². Notably, when considering the same material, laser power density, and duration time, we observe that smaller particles exhibit lower adhesion forces and higher patterning quality. This observation suggests that the size of the particles plays a crucial role in determining the interaction forces and the resulting pattern formation in the given experimental setup. For Fig. R2a3 and b, despite the significant difference in densities and shape among the two types of particles: silica particles (2000 kg/m³) and pollen (1000 kg/m³), the patterning results are similar.

Fig. R2. (appears as S. Fig. 3 of the Supplementary Information) Impact of particle types on pattern formation. a) Comparison of the quality of ring patterns formed by silica particles with diameters of 25 μm , 40 μm , and 55 μm under a laser power density of 1.5 MW/cm² in 3 s. b) Patterns formed by 50 μm pollen particles.

Summary of the modifications:

(1) On Page 3, marked by deep red.

“It can effectively manipulate various particles, such as silica particles ranging from 25 μm to 55 μm in diameter, zirconia particles from 30 μm to 100 μm , pollen particles with a diameter of 50 μm , and so on (detailed in Supplementary Note 2).”

(2) We have added Figs. R2 in the revised Supplementary Information, on Page 7, S. Fig. 3, to clarify the impact of particle types on pattern formation.

3) Finally, the authors make a claim about possible applications in biological systems. Such a claim needs to be slowed down or even removed as it does not demonstrate feasibility or proof of concept for potential use in the biological field. I would suggest that proof of use in biological systems should be provided by testing the assembly of particles, cells or microtissues in liquid media or other biological environments. Alternatively, the authors need to demonstrate how the assembly of particles in air can be used for biological purposes.

We thank the reviewer's comment regarding the need for supporting evidence. To demonstrate the potential applications of our system in biological systems, we conducted preliminary tests involving the manipulation of living cells (shrimp eggs with a diameter of 250 μm) in a water environment. To facilitate this experiment, we introduced a layer of water above the multilayer membrane (Fig. R3a). This arrangement enabled the vibration of the membrane to be transmitted to the water medium. Fig. R3b illustrates the adopted laser pattern, a spiral with a period of 400 μm . Upon illuminating the multilayer membrane with the spiral-patterned laser, an acoustic field in the form of a vortex was excited in the water. This vortex acoustic field induced a counterclockwise rotation of the shrimp egg. The complete rotation motion is depicted in Fig. R3d. The rotated shrimp eggs were able to hatch, indicating that our manipulation did not have any adverse effects on the viability of the eggs. These experimental results provide initial evidence of the system's capability to induce controlled motion of living cells. Further investigations and optimizations can be performed to explore the full potential of this system in biological applications.

Fig. R3: Rotation of a shrimp egg in water using the photoacoustic patterning method. a) Schematic representation of the multilayer membrane setup and water configuration. b) Laser pattern of a spiral with a period of 400 μm . c) Photograph of the Brine shrimp hatched from rotated shrimp egg. d) Sequential images capturing the rotation of a shrimp egg in water at different time intervals.

Summary of the modifications:

(1) On Page 14, marked by deep red.

~~“In the future~~ Furthermore, POAP-PPAP techniques ~~could~~ can be applied in liquid environments, enable the controlled manipulation of targets.”

(2) We have incorporated Figs. R3 in the revised Supplementary Information, on Page 18, S. Fig. 12, to elucidate the possible applications in biological systems.

Reviewer #2 (Remarks to the Author):

In this work, the photoacoustic effects have been used to generate localized Lamb wave fields that can mimic arbitrary laser pattern shapes. By using localized Lamb waves to vibrate the surface of the multilayer membrane, authors can pattern tens of thousands of 25 μ m - diameter silica particles into the desired pattern simultaneously. The results presented in the manuscript are convincing, but the topic is not new, the mechanism explanation is weak, and lack of convincing and exciting application demonstration. Thus, this manuscript in the current version cannot be accepted to be published in nature communications.

We appreciate the reviewer's thorough reading of our work and the valuable feedback provided. We have considered the suggestions and made improvements to our work, which include incorporating theoretical validation and verifying potential applications. Below, we provide a detailed response addressing each of the reviewer's specific questions and comments:

Some questions need to be answered:

1) Since this manuscript is focused on particle control via lamb wave generated by the photoacoustic method, the title “optical-acoustic patterning” is misleading the reader. Optical-acoustic patterning is a kind of means the author wants to pattern the particles via coupling the optical field and acoustic field.

We appreciate the reviewer for bringing up this point, which helps improve the clarity of our manuscript. We recognize the need for clarification and have made the appropriate revision in the manuscript. The term "Optical-acoustic patterning" has been replaced with "photoacoustic patterning" to accurately reflect our focus on particle control using the photoacoustic method.

2) Particle manipulation has been proposed by many published papers, compared with these papers, this manuscript lacks applications. As the author said in the abstract, this work can pave the way for new wave-based manipulation techniques, such as microparticle assembly, biological synthesis, and microsystems, can the author provide at least one of the proposed applications to prove the value of this research?

We appreciate the reviewer for raising this important point. To address this concern, we have conducted additional experiments to validate the potential applications of our

research. We have expanded our system to operate in a liquid environment. As depicted in Fig. R3, we demonstrate the controlled rotation of shrimp eggs using spiral patterns, showcasing the ability to manipulate biological cells. Additionally, as shown in Fig. R4b, we successfully capture and transport silica particles in water, indicating the potential for precise microparticle manipulation. We believe that our system holds potential for microparticle manipulation in liquid environments.

Fig. R4. Illustration of applications in a liquid environment. a) The system's operation in a liquid environment. b) A laser pattern featuring concentric circles with a period of 400 μm. c) Operation of cluster transport of 25 μm silica particles in water. The green circles represent the position of concentric circles, the red dotted circles represent the original position of particles and the red arrows represent the transport direction.

Summary of the modifications:

(1) On Page 14, marked by deep red.

“~~In the future~~ Furthermore, POAP-PPAP techniques ~~could~~ can be applied in liquid environments, enable the controlled manipulation of targets.”

3)What is the resolution for the proposed method and how will the author identify it from the current method?

Thank you for bringing up this question. The resolution of the intaglio pattern we demonstrated in Fig.3c is about 270 μm, which corresponds to the width of the white line in the black-white-black pattern. While, in Fig. 4g, the resolution of the letterpress patterns we demonstrated is about 120 μm, which corresponds to the width of white lines in the letterpress pattern.

A simulation was performed to analyze the vibration of the multilayer membrane under different widths of laser stripes. As shown in Fig. R5a, a wider laser stripe results in stronger vibrations of the multilayer membrane and a wider area of vibration. Specifically, when the width of the laser stripe exceeds 400 μm, the vibrations from both sides of the line overlap, leading to significantly stronger vibrations in the central area around $x = 0$ mm.

By considering the pressure resulting from deformation, the van der Waals forces and the displacement difference between the membrane and particles, we can calculate the

detachment velocity of particles. As a result, we have obtained the spatial distribution of particles detachment velocities, as depicted in Fig. R5b. In Fig. R5c, it is observed that increasing the width of the laser stripe results in a decrease in pattern resolution. It is apparent from Fig. R5d that the positions of the jumping particles on both sides are more localized but less clear when the width of the laser stripe is 150 μm or 300 μm , in comparison with the situation that the width of laser stripe is 400 μm . Furthermore, when the width of the laser stripe is increased to 600 μm , the particles located in the center of the stripe exhibit the ability to jump up to a height of 200 μm . This leads to the formation of a dark pattern in the central region, accompanied by an increase in the width of the pattern.

The calculations are verified by experiments, as depicted in Fig. R5e. Laser stripe with width of 150 μm are less effective in exciting Lamb waves, while laser stripe with width of 600 μm cause multiple peaks and troughs on the membrane, resulting in a decrease in the clarity of the pattern.

Fig. R5. (appears as Fig. S6 of the Supplementary Information) Relationship between patterned resolution and width of laser stripe. a) Spatial distribution of the vibration displacement of the multilayer membrane varies with width of laser stripe. b) Spatial distribution of particles detachment velocities varies with width of laser stripe. c) Relationship between patterned resolution and width of laser stripe. Blue dots represent calculational results, while red dots represent experimental ones. d) Maximal jumping heights of particles at corresponding positions for different laser stripe widths. The light green rectangle represents the laser illumination region. e) Photos of patterned particles with different laser stripe widths at a power density of 1.5 MW/cm².

Summary of the modifications:

(1) On Page 5, marked by deep red.

“Then, we simulated the space-time evolution of vibrations on a multilayer membrane under different widths of laser stripe excitation. Combined with the experimental results, we found that increasing the width of the laser stripe results in a decrease in pattern resolution and an increase of vibration strength, as shown in supplementary Note 4. Additionally, when the width of the laser stripe exceeds 430 μm , it results in vibration overlap, while a width below 280 μm leads to insufficient vibration excitation, ultimately affecting the clarity of the patterns. The width of 400 μm is proven to be an optimal value and corresponds a pattern resolution of 270 μm .”

(2) We have added Figs. R5 in the revised Supplementary Information, on Page 11, S. Fig. 6, to clarify the resolution of PPAP and the impact of the width of laser stripe.

4) For the mechanism discussion, how did the 25 μm particle was chosen to run the experiment? Can the author use other particles with different diameters to demonstrate the control mechanism?

We express our gratitude to the reviewer for raising this significant question. we aimed to choose a particle size that allowed us to observe clear and distinguishable manipulation effects. As the particle diameters decrease, the influence of the Van der Waals force becomes notably pronounced. To explore this phenomenon further, we conducted experiments using particles with different sizes and materials (as depicted in Fig. R6). Through our investigations, we found that 25 μm silica particles yielded the clearest experimental results.

Fig. R6. (appears as Fig. S3 of the Supplementary Information) Pattern of particles with different diameters and materials. a) Ring patterns formed by silica particles with diameters of 25 μm , 40 μm , and 55 μm under the laser irradiation of 1.5 MW/cm² power density in 3 s. b) The patterns formed by zirconia particles with diameters of 30 μm , 50 μm , and 100 μm .

Summary of the modifications:

(1) On Page 3, marked by deep red.

“It can effectively manipulate kinds of particles, such as silica particles ranging from 25 μm to 55 μm in diameter, zirconia particles from 30 μm to 100 μm , pollen particles with a diameter of 50 μm , and so on (detailed in Supplementary Note 2).”

(2) We have incorporated Figs. R2 in the revised Supplementary Information, on Page 7, Fig. S3, to Impact of particle types on pattern formation.

5) Figure 3 and Figure 4 almost deliver a similar message, one does not see the technology, mechanism, or application as different.

We sincerely appreciate the reviewer for raising this question and bringing these concerns to our attention. It has helped us identify that there are still misleading aspects in our description of Fig. 4. The patterns as shown in Fig. 4, which is an extension of Fig. 3, showcase the enhanced complexity and superiority of our DMD programmable method. During the process of the intaglio style pattern, the laser pattern remains unchanged. Therefore, it is not possible to remove particles from the background. This can be observed in Fig. R7a and b, where one frame corresponds to a distinct pattern.

Fig. R7. (appears as Fig. 4 of the Main text) Difference between Fig. 3 and Fig. 4 in the main text. a) The intaglio style target image. **b)** The resulting intaglio-style pattern of pig'sy. **c)** The letterpress style target image. **d)** The resulting letterpress-style pattern of a flower. **e)** Three sequential frames of the outer area pattern. Blur lines indicate the evolving trajectory of the innermost lines of the first frame. **f)** The process of removing particles from the outer area of the flower. **g)** Three sequential frames of the inner area pattern. **h)** The process of driving particles in the inner area of the flower.

In contrast, for letterpress style pattern, Fig. R7c-g exemplify how the 1152 laser pattern frames continuously evolve over time during the formation process, leading to the effective removal of particles from the background on a large scale. Although this method requires more time to complete, it selectively retains only the particles that constitute the desired shape on the multilayer membrane surface, resulting in a precise and accurate pattern formation.

Summary of the modifications:

(1) On Page 11, marked by deep red, we've revised the Fig. 4.

~~“To achieve this the result shown in Fig. 4b, the green area we need to be break down the green area into numerous closed curves with identical width and spacing, as shown in Fig. 4b. This can be viewed as a periodic grating with a period of 0.8 mm and a 50% duty cycle. The particles tend to gather near the center of the stripe. Therefore, we can manipulate the particles on the multilayer film by moving the laser stripe (for detailed information, refer to Supplementary Note 5).”~~

~~“To achieve the letterpress style pattern of a kapok flower as shown in Fig. 4b, dynamic playback of 1152 frames of images at a frame rate of 10 fps is required using the DMD. The laser pulses emitted by the laser system are synchronized with the pattern sequence. As a result, the laser pattern projected onto the multilayer film changes at a rate of 10 fps. The design process of the image sequence is depicted in Supplementary Note 6. The first 576 frames are used to remove particles from the external background of the kapok flower pattern, while the remaining 576 frames are used to transport particles from inner to its contours.”~~

~~“Similar to the POAP process utilized for the Pigsy pattern, The membrane is initially coated uniformly with the silica particles as above mentioned. The letterpress patterning procedure consists of two steps. Fig. 4c illustrates the first step 576 frames, where the curves of the outer area sequentially expand outward, and the PPAP removes particles located in the outer area (see Supplementary Video 2 and Note 6). The picture of the particles driven by the PPAP after 8.4 seconds of the initial laser pulse are shown in Fig. 4d, where particles outside of the flower are gradually eliminated. After 57.6 seconds, the particles in the outer region are almost removed.”~~

~~“The second step remaining 576 frames are shown in Fig. 4de, where the inner region is decomposed into multiple connected domains, each of which expands outward independently. The picture of the particles driven by the POAP 8.4 seconds after the initial laser pulse are seen in Fig. 4e, where particles outside of the flower are gradually eliminated. After six periods of the first step as shown in Fig. 4e, the particles in the outer region are almost removed. Fig. 4f displays the state of the patterning process after 64.8 seconds of the initial laser pulse after the laser pulse. The internal region is partitioned into various interconnected domains by curves, and the particle ripples are clearly visible. The final result, obtained at 115.2 seconds, is shown in Fig. 4g, where the uniform layer of particles has been successfully transformed into a kapok flower, closely matching the target image in Fig. 4a.”~~

6)The author provides that “the Lamb wave transfers kinetic energy from the membrane to the particles.” One believes that should be true, this experiment

phenomenon is kind of like the Chladni plate experiment, the author may provide a more accurate and detailed theoretical explanation to support the experimental result.

We appreciate the reviewer's suggestion for a more accurate and detailed theoretical explanation to support our experimental results. Indeed, we agree with the reviewer's observation that our experimental phenomenon shares certain similarities with the Chladni plate experiment. The Chladni plate is excited through a vibrating source, resulting in the generation of eigenmode vibrations on the plate. The arrangement of particles on the plate is constrained by the plate's modes, which are determined by its geometry and material properties.

In our approach, we use a laser to excite the multilayer membrane, which generates localized vibrations capable of driving the particles. By utilizing the DMD-reflected laser stripes, we are able to dynamically manipulate the particles on the multilayer membrane in a programmable manner, making the localized manipulation more flexible compared with the Chladni plate method.

The programmable photoacoustic patterning method involves the sequential unfolding of the physical processes as shown in Fig. R8a.

- 1) The multilayer membrane absorbs the energy from the laser, resulting in a temperature increase.
- 2) The temperature rise induces thermal expansion, leading to the propagation of Lamb waves in the multilayer membrane.
- 3) The multilayer membrane impacts the particles, causing them to be dislodged.

Given that the jumping time of the particles is 10 ms, which is much shorter than the pulse interval of the laser (100 ms). This allows for the analysis of the specific physical phenomena under the condition of individual pulse excitation.

Fig. R8b shows the spatial-temporal distribution of vibrations when the width of the laser stripe is 400 μm and the laser power density is 2 MW/cm^2 .

We conducted a statistical analysis of the spatial distribution of the maximum velocities when the displacements and velocities of the multilayer membrane are positive. The results are represented by the blue curve in Fig. R8c. Due to the influence of van der Waals force and deformation pressure, the velocities of particles decrease during the detachment process (purple curve in Fig. R8c). Particles with lower velocities experience more deceleration during this process.

Using the detachment velocities of particles, we can calculate the maximum jumping height of the particles, considering the effects of gravity and air resistance. The theoretical results, represented by the blue curve in Fig. R8d, align with the experimental ones depicted by the red dots. The width of the laser stripe has a notable impact on the results, as shown in Fig. R8e for widths of 300 μm and 600 μm .

The velocities of particles after the detachment stage under different laser power densities are shown in Fig. R8g. The corresponding changes in the jumping height of particles can be observed by the green semicircular dots in Fig. R8h. The calculation results slightly differ from the experimental results represented by the blue dots, which may be due to the lack of consideration for camera's measurement error and particle rotation during the jumping process.

Fig. R8. Theoretical explanation about the programmable photoacoustic patterning method.

a) The evolutions of the laser power density, transient temperature rise, displacement of the multilayer membrane, and particle motions over time. b) Calculational results depicting space-time diagrams of elastic waves. c) Velocities of particles before and after the detachment stage of corresponding positions. d) Maximum jumping heights of particles of corresponding positions. Blue curve represents calculation results, while red dots represent experimental results. The light green rectangle represents the laser illumination region. e) Maximum jumping heights of particles of corresponding positions, while the widths of laser stripes are 300 μm and 600 μm respectively. f) Spatial distribution of the vibration displacement of the multilayer membrane varies with changes in laser power density. g) Spatial distribution of detachment velocities of particles varies with changes in laser power density. h) Relationship between particles' jumping heights and laser power densities. The green semicircles represent the calculation results, while the blue circles represent the experimental results.

Summary of the modifications:

(1) On Page 7, marked by deep red, we've revised the Fig. 2d, e.

“The main force that hinders the separation of particles from the membrane is the van der Waals force, which is tens of thousands of times stronger than the gravitational force acting on the particles. Based on the supplementary Note 3 and Note 4, we can calculate the spatial distribution of the maximum jumping height of particles. The calculated

results with a laser power density of 2 MW/cm² are depicted by blue curve in Fig. 2d, demonstrating the agreement with the experimental results represented by red dots. The jumping heights of particles vary depending on their positions relative to the laser-illuminated region. The sparse distribution of particles decreases the interactions between them, facilitating their return to the original positions after jumping. If a thick layer of particles is spread over the multilayer membrane, the particles will collide and scatter in various directions. As a result, the particles within the irradiation area will be rearranged a few seconds after the initial laser pulses.”

~~“Particles on the membrane jump and are displaced by the elastic waves. Fig. 2e demonstrates that the jumping height of the particles increases with the power of the laser. The laser power significantly determines the strength of vibration, which, in turn, influences the jumping height of particles and the final arrangement of particles. Based on the information about vibrations from the simulation results using different laser power density, we calculated the maximum jumping height of particles using numerical integration methods. The results, represented by the green semicircles in Fig. 2e, exhibit slight differences compared to the experimental results represented by the blue dots. This discrepancy may be attributed to the omission of camera’s measurement error and particle rotation during the jumping process in the calculations. Both experimental and calculated results demonstrate that the minimum laser power density required for particles to jump is 1.2 MW/cm². once the power density exceeds the threshold of 1 MW/cm². The inset of Fig. 2d shows the projectile motion of a particle captured by a high-speed camera, with a maximum height of 60 μm.”~~

~~“As shown in Fig. 2e, the jumping heights of particles vary depending on their positions relative to the laser-illuminated region. The sparse distribution of particles decreases the interactions between them, facilitating their return to the original positions after jumping. If a thick layer of particles is spread over the multilayer membrane, the particles will collide and scatter in various directions. As a result, the particles within the irradiation area will be rearranged a few seconds after the initial laser pulses.”~~

Reviewer #3 (Remarks to the Author):

Comments: In this manuscript, the author used photoacoustic effects to generate localized Lamb wave fields that can mimic arbitrary laser pattern shapes. By using localized Lamb waves to vibrate the surface of the multilayer membrane membrane, they can pattern tens of thousands of 25μm in-diameter silica particles into the desired pattern simultaneously.

The manuscript is well-organized, and this process is good and may attract broad interest in the field of acoustics. However, several issues still need to be addressed before consideration for publication.

We would like to express our gratitude to the reviewer for their positive feedback on our manuscript and for their support in its publication. We have prepared a detailed

point-by-point response to the referees, along with the necessary revisions made in the manuscript. Please find the attached document containing these responses.

1) It is mentioned in the manuscript that POAP technology can realize the dynamic flow of particles by quickly and continuously adjusting the laser shape, thereby creating frame-by-frame animation effects. The authors should explain and analyze the mechanism of this active flow in detail to ensure that the particles can flow accurately along the corresponding elastic wave field.

We appreciate the reviewer for bringing the unclear statements in our manuscript to our attention. To provide a clear explanation of the dynamic flows, we conducted simulations considering gravitational force, air friction, and particle collision. In Fig. R9, we simulate the behavior of 20 silica particles with diameter of 25 μm , spanning from 0.08 mm to 1.6 mm. The laser stripe, with a width of 400 μm , moves from $x = 0$ mm to $x = 4$ mm over a 16 seconds duration in one cycle.

The temporal-spatial positions of the 20 particles in the X direction were recorded every 4 seconds, as depicted in Fig. R9. The color shading at each data point indicates the number of particles at that position. It can be observed that after 8 cycles (128 seconds), the particles tended to move towards positions near $x = 4$ mm.

This simulation provides an explanation for the dynamic flow of particles driven by a laser stripe. Thus, by moving the laser stripe, we can effectively manipulate the particles on the membrane.

Additionally, in Fig. 4 in main text, we demonstrate that periodic lines are more efficient than a single line, facilitating the sieving of particles to the desired location.

Fig. R9. (appears as S. Fig. 7 of the Supplementary Information) Simulation of dynamic flow driving by laser strip. The spatial position of particles changes as the laser stripe moves. Red dots represent particles, and the shade of color represents the number of particles overlaps at that position.

Summary of the modifications:

(1) On Page 11, marked by deep red.

“The particles tend to gather near the center of the stripe. Therefore, we can manipulate the particles on the multilayer film by moving the laser stripe (for detailed information, refer to Supplementary Note 5).”

(2) We have added Figs. R8 in the revised Supplementary Information, on Page 12, S. Fig. 7, to clarify the process of dynamic flow.

2)The authors mentioned that POAP technology can efficiently manipulate particles on multi-layer membranes through the photoacoustic effect and Lamb wave in the air. The authors should conduct a detailed study and analysis of the mechanical properties during this manipulation. For example, how does the laser's power and frequency affect particle manipulation?

We appreciate the reviewer for bringing up this question. We analyzed the mechanical properties using simulations and experiments. The simulations focused on studying the vibration of the multilayer membrane under different laser power densities. As shown in Fig. R10a, there is a positive correlation between the laser power density and the vibration displacement of the multilayer membrane. Additionally, we simulated the jumping height of particles under different laser power densities.

Fig. R10. (appears as S. Fig. 5 of the Supplementary Information) Impact of laser power density on the quality of particle patterning. a) Spatial distribution of the vibration displacement of the multilayer membrane varies with changes in laser power density. b) Spatial distribution of particles detachment velocities varies with changes in laser power density. c) Relationship between particles' jumping height and laser power density. The green dots represent the calculation results, while the blue dots represent the experimental results. d) Comparison of particle patterning quality under different laser power densities.

Considering factors such as deformation pressure, van der Waals forces, and displacement difference between the membrane and particles, we calculated the detachment velocities of particles. The spatial distribution of velocities is shown in Fig.

R10b.

Using the particle's detachment velocity, the maximum jumping height of the particle can be calculated. The calculation results represented by green semicircles slightly differ from the experimental results represented by the blue dots, which may be due to the lack of consideration for camera's measurement error and particle rotation during the jumping process.

This conclusion is further supported by experimental results, as illustrated in Fig. R10d. The figure displays the arrangement of circular patterns with a size of 1 cm under different laser powers. It is evident from the figure that higher laser power results in better arrangement outcomes.

Summary of the modifications:

(1) On Page 8, marked by blue, we've revised the Fig. 1 (c)

~~“Particles on the membrane jump and are displaced by the elastic waves. Fig. 2e demonstrates that the jumping height of the particles increases with the power of the laser. The laser power significantly determines the strength of vibration, which, in turn, influences the jumping height of particles and the final arrangement of particles. Based on the information about vibrations from the simulation results using different laser power density, we calculated the maximum jumping height of particles using numerical integration methods. The results, represented by the green semicircles in Fig. 2e, exhibit slight differences compared to the experimental results represented by the blue dots. This discrepancy may be attributed to the omission of camera's measurement error and particle rotation during the jumping process in the calculations. Both experimental and calculated results demonstrate that the minimum laser power density required for particles to jump is 1.2 MW/cm². once the power density exceeds the threshold of 1 MW/cm². The inset of Fig. 2d shows the projectile motion of a particle captured by a high-speed camera, with a maximum height of 60 μm.”~~

(2) We have added Figs. R9 in the revised Supplementary Information, on Page 10, S. Fig. 5, to clarify the impact of laser power density on the quality of particle patterning.

3)The authors claimed that POAP technology can pattern many particles into the desired pattern. Did the authors evaluate the precision and reproducibility of this modeling procedure? Are there any errors or deviations? If so, have the authors proposed methods or measures for improvement?

We appreciate the reviewer for bringing up this issue. The resolution of the intaglio we presented is 200 μm, while the resolution of the letterpress patterns is 100 μm. The precision of our system is primarily determined by the width of the laser stripes. As observed from Fig. R11a, wider laser stripe induces stronger vibration of the multilayer membrane and wider vibration area. Particularly, when the width of the laser stripe exceeds 400 μm, the vibrations on both sides of the line overlap, resulting in significantly stronger vibrations in the central area around $x=0$ mm. By applying the

previously mentioned calculation method, we obtained the spatial distribution of particle detachment velocity, as shown in Fig. R11b. In Fig. R11c, it is observed that increasing the width of the laser stripe results in a decrease in pattern resolution.

It is apparent from Fig. R11d that the positions of the jumping particles on both sides are more localized but less clear when the width of the laser stripe is 150 μm or 300 μm , in comparison with the situation that the width of laser stripe is 400 μm . Furthermore, when the width of the laser stripe is increased to 600 μm , the particles located in the center of the stripe exhibit the ability to jump up to a height of 200 μm . This leads to the formation of a dark pattern in the central region, accompanied by an increase in the width of the pattern.

The calculations are verified by experiments, as depicted in Fig. R11e. Laser stripe with width of 150 μm is less effective in exciting Lamb waves, while laser stripe with width of 600 μm cause multiple peaks and troughs on the membrane, leading to a decrease in the clarity of the pattern.

Fig. R11. (appears as S. Fig. 6 of the Supplementary Information) Relationship between patterned resolution and width of laser stripe. a) Spatial distribution of the vibration displacement of the multilayer membrane varies with changes in width of laser stripe. b) Spatial distribution of particles detachment velocities varies with changes in width of laser stripe. c) Relationship between patterned resolution and width of laser stripe. Blue dots represent calculation results, while red dots represent experiment results. d) maximum jumping heights of particles at corresponding positions are compared for laser stripe widths. The light green rectangle area represents the laser illuminated region. e) Comparison of patterned effects is shown with different laser stripe widths at a laser power density of 1.5 MW/cm².

The method shows good repeatability, as depicted in Fig. R12. The figure displays two distinct patterns, each replicated twice, demonstrating consistent patterning outcomes. Factors such as optical aberration, power density, and beam uniformity can affect the imaging quality. To enhance the quality of programmable photoacoustic patterning, it is advisable to use an optical system with low aberration and laser equipment with superior uniformity.

Fig. R12. Demonstration of the high repeatability of the method. a) Two instances of particle patterning results for a calm piggy pattern. b) Two instances of particle patterning results for a smile piggy pattern. These results illustrate the consistent outcomes achieved by our method.

Summary of the modifications:

(1) On Page 5, marked by deep red.

“Then, we simulated the space-time evolution of vibrations on a multilayer membrane under different widths of laser stripe excitation. Combined with the experimental results, we found that increasing the width of the laser stripe results in a decrease in pattern resolution and an increase of vibration strength, as shown in supplementary Note 4. Additionally, when the width of the laser stripe exceeds 430 μm , it results in vibration overlap, while a width below 280 μm leads to insufficient vibration excitation, ultimately affecting the clarity of the patterns. The width of 400 μm is proven to be a optimal value and corresponds a pattern resolution of 270 μm .”

(2) On Page 13, marked by deep red.

“Additionally, increased resolution and quality may be achieved by improving optical system with low aberration and laser equipment with superior uniformity. ~~through reducing the multilayer membrane thickness and laser pulse width.~~”

(3) We have added Figs. R11 in the revised Supplementary Information, on Page 11, S.

Fig. 6, to clarify the resolution of PPAP and the impact of changes in the width of the laser stripe.

4)It is mentioned in the manuscript that POAP technology can be combined with closed-loop control and layer-by-layer transfer to realize a revolution in the field of micro/nano manufacturing. May I ask if the authors have explored or verified the practical application of this combination? Has it achieved practical application results in the fields of particle assembly, biosynthesis, or microsystems?

We greatly appreciate the valuable concerns raised by the reviewer, and we will certainly consider them when validating the potential applications of this method in our future work. The schematic of layer-by-layer transferring is shown in Fig. R13a. We first patterned the quartz sands with size of 150 μm on the multilayer membrane.

Secondly, we brought a base plate, coated with a layer of polymer adhesive, into contact with the front side of the intaglio-patterned sands. Finally, the patterned sands on the multilayer were then transferred onto the base plate. By repeating the aforementioned process, the particles were stacked layer by layer, forming a 3D printed object.

The printing results are shown in Fig. R13b and c. For the structure of "I," consists of three layers. On the other hand, the structure of "T" consists of five layers. Our method offers the flexibility to choose the material, shape, and size of the particles for each layer.

While the initial results presented in Fig. R13 may exhibit some discrepancies compared to commercial 3D printing systems, we have confidence that this method can be further refined to achieve a level suitable for practical applications in the future.

Fig. R13. (appears as Fig. S11 of the Supplementary Information) Layer-by-layer 3D printing. a) Schematic of the photoacoustic 3D printing process. b) Top view of the 3D printed object. c) Front view of the 3D printed object.

Summary of the modifications:

(1) On Page 14, marked by deep red.

“These patterned particles may be transferred layer-by-layer to create a 3D structure through micro 3D printing (detailed in supplementary Note 9).”

(2) We have added Figs. R12 in the revised Supplementary Information, on Page 12, S. Fig. 11, to clarify the process of layer-by-layer 3D printing.

5)The authors mentioned that POAP technology can be applied in the liquid environment to realize the control and manipulation of the target in the liquid. May I ask which fluid environment the author chooses for the experiment and whether there is a big difference in the experimental results of different fluid environments?

We thank the reviewer's insightful comments, and we acknowledge the critical questions raised, which will guide our future work. In response to these concerns, we have expanded our system to operate in a liquid environment, as shown in Fig. R14a. By utilizing concentric circular laser patterns, as depicted in Fig. R14b, we successfully captured and transported silica particles in water. In an air environment, the particles

are mainly influenced by the adhesive force and pressure exerted by the multilayer membrane. Alternatively, when operating in a fluid environment, the particles are primarily affected by the acoustic radiation force and the viscous force. The acoustic radiation force-based manipulation method has excellent biocompatibility. We applied a spiral laser pattern, as depicted in Fig. R3b, to manipulate living cells. Fig. R3c demonstrates the controlled rotation of shrimp eggs achieved through this method. It provides a clear advantage over previously published methods.

Fig. R14. Illustration of applications in a liquid environment. a) The system's operation in a liquid environment. b) A laser pattern featuring concentric circles with a period of 400 μm . c) Operation of cluster transport of 25 μm silica particles in water. The green circles represent the position of concentric circles, the red dotted circles represent the original position of particles and the red arrows represent the transport direction.

6) There are many grammatical and writing errors in the manuscript. Please correct them before publication.

We thank the reviewer for the careful reading of our manuscript. We have revised the manuscript.

Reviewers' Comments:

Reviewer #1:

Remarks to the Author:

The author has successfully responded to the reviewers' comments. I believe that the quality of the manuscript has now improved sufficiently for me to propose it for publication in this journal.

Reviewer #2:

Remarks to the Author:

The author has effectively addressed the majority of queries in the response, leading to substantial improvements in the manuscript based on the reviewer's comments. However, there remains a single question that lacks clarity.

The author endeavored to elucidate the mechanism by offering additional details from their experiment. While other reviewers have noted a similar topic previously reported, this paper surpasses the quality of the published work. However, it may not meet the elevated standards of this journal. Clarifying the mechanism is crucial as an innovative aspect. Readers may appreciate a more in-depth exploration, including theoretical derivations illustrating the control of particles. Specifically, insights into how laser energy transforms to the plate, as indicated by equation (3) in the supplementary material, and the process by which particle force is derived from the mentioned equation, would enhance the paper. Simply presenting the governing equation (3) is undoubtedly insufficient to thoroughly elucidate the mechanism.

Therefore, it is imperative to enhance this crucial aspect to achieve the level of innovation required for the acceptance and publication of this paper in Nature Communications.

Reviewer #3:

Remarks to the Author:

The author has successfully responded to the reviewers' comments. I believe that the manuscript's quality has improved sufficiently for me to propose it for publication in this journal.

Reviewer #1 (Remarks to the Author):

The author has successfully responded to the reviewers' comments. I believe that the quality of the manuscript has now improved sufficiently for me to propose it for publication in this journal.

We would like to express our gratitude to the reviewer for their positive feedback on our manuscript and for their support in its publication.

Reviewer #2 (Remarks to the Author):

The author has effectively addressed the majority of queries in the response, leading to substantial improvements in the manuscript based on the reviewer's comments. However, there remains a single question that lacks clarity.

The author endeavored to elucidate the mechanism by offering additional details from their experiment. While other reviewers have noted a similar topic previously reported, this paper surpasses the quality of the published work. However, it may not meet the elevated standards of this journal. Clarifying the mechanism is crucial as an innovative aspect. Readers may appreciate a more in-depth exploration, including theoretical derivations illustrating the control of particles. Specifically, insights into how laser energy transforms to the plate, as indicated by equation (3) in the supplementary material, and the process by which particle force is derived from the mentioned equation, would enhance the paper. Simply presenting the governing equation (3) is undoubtedly insufficient to thoroughly elucidate the mechanism.

Therefore, it is imperative to enhance this crucial aspect to achieve the level of innovation required for the acceptance and publication of this paper in Nature Communications.

We would like to express our gratitude to the reviewer for their positive evaluation and insightful comments on our manuscript. Taking into consideration the reviewers' feedback, we have made significant improvements to the theoretical derivation presented in our manuscript, specifically emphasizing the interplay between light, heat, and vibration phenomena. We have provided a more comprehensive analysis of the impact of laser power density on both the temperature and displacement of the multilayer membrane. Additionally, we have incorporated numerical integration methods to accurately calculate the forces experienced by silica particles positioned on the multilayer membrane.

We commenced the derivation by analyzing the effects of laser irradiation and subsequently deduced the opto-thermal process and heat diffusion within the multilayer membrane. The spatial and temporal temperature distribution within the multilayer membrane was determined and presented. Using the strain induced by temperature as the initial condition, we calculated the initial displacement and illustrated the spatiotemporal vibration distribution in the metallic membrane. The derivation process of the equations is outlined in Fig. R1.

Fig. R1. (appears as S. Fig. 1 of the Supplementary Information) **Flowchart of the derivation process.**

The numerical values of the physical quantities used in the derivation are summarized in Table R1.

name	symbol	value	name	symbol	value
Reflectivity	R	0.41	Poisson ratio	ν	0.3
Thermal capacity	C_p	$500 J/Kg \cdot m$	Young's modulus	E	210 GPa
Pulse width	τ	6 ns	Membrane thickness	d	$5 \mu m$
Optical absorption coefficient	α	$7 \times 10^7 m^{-1}$	Sound velocity in water	c_f	1490 m/s
Illumination width	2L	$400 \mu m$	Water density	ρ_f	$1000 Kg/m^3$
Thermal conductivity	κ	$45 W/m \cdot K$	Temperature dissipation distance	l	90 nm
Stainless-steel density	ρ	$7930 Kg/m^3$	viscosity ratio of water	η	$1 \times 10^{-3} Pa \cdot s$

Table R1. Numerical values of physical quantities

We assume that the basic illumination unit is a $2L = 400 \mu m$ wide line has a constant laser power in the Y direction. Therefore, we can express the laser power as follows:

$$I(z, t) = \begin{cases} I_0(1 - R)e^{-\frac{4(t-\tau)^2}{\tau^2}} e^{-\alpha z} & (|x| < L) \\ 0 & (|x| \geq L) \end{cases} \quad (1)$$

where I is the laser power density, I_0 is the initial laser power density, R is the reflectivity of materials, τ is the pulse width, α is the optical absorption coefficient, z is the penetration depth of laser. We simplified the multilayer membrane as a $5 \mu m$ stainless-steel material with a light energy absorption ratio equivalent to that of TiN.

We can calculate the temperature spatial-time distribution in multilayer membrane through the thermal conduction equation expressed as follows, where Q represents the input heat, which in this section is supplied by the absorbing laser energy [1].

$$\rho C_p \frac{\partial T}{\partial t} - \nabla \cdot (\kappa \cdot \nabla T) = Q \quad (2)$$

where ρ is the density, C_p is the thermal capacity and κ is the thermal conductivity of stainless-steel. The laser light source is uniform in the x-direction within the region of $|x| < L$, meaning that the resulting temperature is also uniformly distributed in the x-direction within that region. Therefore, we can rewrite the above equation into the following one-dimensional form:

$$\rho C_p \frac{\partial T}{\partial t} - \kappa \frac{\partial^2 T}{\partial z^2} = \begin{cases} \alpha I_0 (1 - R) e^{-\frac{4(t-\tau)^2}{\tau^2}} e^{-az} & (|x| < L) \\ 0 & (|x| \geq L) \end{cases} \quad (3)$$

Given the high optical absorption coefficient of stainless steel, the laser penetration depth in the membrane is shallow. As a result, we can assume that only the surface absorbs the laser energy, leading to a localized increase in temperature that subsequently diffuses into the interior of the multilayer membrane. Equation (3) can be reformulated as follows:

$$\rho C_p \frac{dT}{dt} = \alpha I_0 (1 - R) e^{-\frac{4(t-\tau)^2}{\tau^2}} + \kappa \frac{T}{l^2} \quad (|x| < L) \quad (4)$$

where l is the temperature dissipation distance. By solving the above differential equation, we can obtain the variation of the surface temperature of stainless steel with respect to time:

$$T(0, t) = \frac{\tau \sqrt{\pi} \alpha I_0 (1 - R)}{4a} e^{A^2} e^{\frac{\kappa(\tau-t)}{al^2}} \left[\operatorname{erf}\left(2 + A\right) - \operatorname{erf}\left(2 + A - \frac{2t}{\tau}\right) \right] \quad (5)$$

where $a = \rho C_p$, $A = \frac{\tau \kappa}{4al^2}$ and $\operatorname{erf}(X)$ is error function. At this point, the problem transforms into a classical heat diffusion problem with temperature boundary conditions:

$$\begin{cases} \rho C_p \frac{\partial T}{\partial t} - \kappa_z \frac{\partial^2 T}{\partial z^2} = 0 \\ T(z, 0) = 0 \\ T(0, t) = I_0 f(t) \end{cases} \quad (6)$$

where $f(t) = \frac{\tau\sqrt{\pi}\alpha(1-R)}{4a} e^{A^2} e^{-\frac{\kappa(\tau-t)}{al^2}} [\text{erf}(2+A) - \text{erf}(2+A - \frac{2t}{\tau})]$.

The solution takes the following form:

$$T(z, t) = I_0 \int_0^t f'(h) \cdot \text{erfc}\left(-\frac{z\sqrt{\rho C_p}}{2\sqrt{\kappa_z(t-h)}}\right) \frac{dh}{\sqrt{h}} \quad (7)$$

where $\text{erfc}(X)$ is co-error function. We conducted a comparison between the spatiotemporal distribution of the temperature field obtained from our solution and the simulated results obtained using COMSOL. The comparison is presented in Fig. R2a and R2b. In this comparison, the laser power density used is 1 MW/cm^2 .

Fig. R2. (appears as S. Fig. 2 of the Supplementary Information) **Space-time distribution of the temperature field obtained from the theoretical solution and the simulated results.** a) Theoretical results depicting space-time diagrams of temperature induced by the photothermal effect. b) Simulation results depicting space-time diagrams of temperature induced by the photothermal effect.

Based on the aforementioned figure, it is evident that the temperature variation primarily occurs on the surface of the multilayer membrane and dissipates rapidly. This observation provides valuable insights for deriving the vibration field of the metallic membrane. The equation of motion in a solid medium can be expressed as follows [2]:

$$\rho \frac{\partial^2 u}{\partial t^2} = \nabla \cdot \sigma + F \quad (8)$$

where u is the displacement, σ is stress and F is external force which oriented from thermal expansion in this section. As the rapid temperature change occurs within a brief duration of 10 ns, which is significantly shorter compared to the vibration dynamics, we can treat the effects of thermal expansion as an initial condition rather than incorporating external forces into the differential equation. Considering the specific nature of the system in this paper, we can make certain approximations. Since the thickness of the multilayer membrane is much smaller than the possible wavelength of vibrations, and its width is much larger than the possible wavelength of vibrations, the entire system can be approximated as the vibration of an infinitely long slender beam under impact. The vibration equation governing the transient behavior of an infinitely long slender beam can be expressed as follows:

$$\frac{\partial^4 u_z}{\partial x^4} + \frac{1}{c^2 K^2} \frac{\partial^2 u_z}{\partial t^2} = 0 \quad (9)$$

where u_z is the displacement in Z direction, $c = \sqrt{\frac{E}{\rho}}$ is the longitudinal wave velocity, E is Yang's modulus and $K = \frac{d}{\sqrt{12}}$ is the sectional radius of gyration of a rectangular shape with a thickness of d . The solution of equation (9) is

$$u_z(x, t) = \int_{-\infty}^{+\infty} e^{i\beta x} [B(\beta) \cos(\omega t) + D(\beta) \sin(\omega t)] d\beta \quad (10)$$

where ω is angular frequency, $\beta = \sqrt{\frac{\omega}{cK}}$ is spatial frequency, $B(\beta) = \frac{1}{2\pi} \int_{-\infty}^{+\infty} e^{-i\beta x} u(x, 0) dx$ and $D(\beta) = \frac{1}{2\pi\omega} \int_{-\infty}^{+\infty} e^{-i\beta x} v(x, 0) dx$. $u(x, 0)$ and $v(x, 0)$ are initial displacement and velocity respectively. At this stage, it is necessary to determine the initial motion state of the multilayer membrane. Considering that the expansion of an object due to heating corresponds to a change in strain, equation (8) should be reformulated as follows [3]:

$$\rho \frac{\partial^2 u}{\partial t^2} = \nabla \cdot [C : (\varepsilon - \gamma T)] \quad (11)$$

where C is elasticity tensor, ε is strain tensors, γ is the thermal expansion tensor. Let's rearrange equation (11) in terms of displacement as the variable:

$$\left\{ \begin{array}{l} \rho \frac{\partial^2 u_x}{\partial t^2} = \frac{E(1-\nu)}{(1+\nu)(1-2\nu)} \frac{\partial^2 u_x}{\partial x^2} + \frac{E \left(\frac{\nu}{1-2\nu} + \frac{1}{4} \right)}{(1+\nu)} \frac{\partial^2 u_z}{\partial x \partial z} + \dots \\ \quad \frac{E}{4(1-\nu)} \frac{\partial^2 u_x}{\partial z^2} - \frac{E}{1+\nu(1-2\nu)} \frac{\partial \gamma T}{\partial x} \\ \rho \frac{\partial^2 u_z}{\partial t^2} = \frac{E(1-\nu)}{(1+\nu)(1-2\nu)} \frac{\partial^2 u_z}{\partial z^2} + \frac{E \left(\frac{\nu}{1-2\nu} + \frac{1}{4} \right)}{(1+\nu)} \frac{\partial^2 u_x}{\partial x \partial z} + \dots \\ \quad \frac{E}{4(1-\nu)} \frac{\partial^2 u_z}{\partial x^2} - \frac{E}{(1+\nu)(1-2\nu)} \frac{\partial \gamma T}{\partial z} \end{array} \right. \quad (12)$$

where ν is Poisson ratio. Since u_x is much smaller than u_z , we neglect the effect of u_x here. The equation is rearranged as follows:

$$\rho \frac{\partial^2 u_z}{\partial t^2} = \frac{E(1-\nu)}{(1+\nu)(1-2\nu)} \frac{\partial^2 u_z}{\partial z^2} - \frac{E}{(1+\nu)(1-2\nu)} \frac{\partial \gamma T}{\partial z} \quad (13)$$

Solving the above equation yields:

$$u_z = \frac{\partial \gamma T}{\partial z} \frac{d^2}{1-\nu} \left(1 - \exp \left(- \sqrt{\frac{E}{\rho(1+\nu)(1-2\nu)}} \frac{t}{d} \right) \right) \quad (14)$$

The spatial-temporal distribution of displacement obtained does not account for the propagation of vibrations in the x-direction, thus it remains relatively accurate only for the initial tens of nanoseconds. To address this, we can utilize the maximum displacement from the distribution as the initial displacement function and substitute it into equation (10):

$$u_z(x, t) = \frac{\max(u_z)}{\pi} \int_{-\infty}^{+\infty} e^{i\beta x} \frac{\sin(L\beta)}{\beta} \cos(\omega t) d\beta \quad (15)$$

In the experiment, there exists a water layer beneath the multilayer membrane, with an approximate thickness of 2 cm. The presence of this water layer introduces impedance to the vibration, and the level of impedance is dependent on the frequency of the vibration. There are two main forms of attenuation for Lamb waves in water: viscous

attenuation δ_L and leaky wave attenuation δ_v , The attenuation coefficient is given by:

$$\delta_L = \frac{2\omega^2\eta}{3c_f^3\rho_f} \quad (16)$$

$$\delta_v = \frac{1}{2d} \sqrt{\frac{\omega\eta\rho_f}{2E\rho_s}} \quad (17)$$

where η is viscosity ratio and c_f is velocity and ρ_f is density of fluid. Taking into account the attenuation of lamb wave, the displacement equation is given by:

$$u_z(x, t) = \frac{\max(u_z)}{\pi} \int_{-\infty}^{+\infty} e^{i\beta x} \frac{\sin(L\beta)}{\beta} \cos(\omega t) \cdot 10^{-\left[\frac{(\delta_L+\delta_v)x}{20}\right]} d\beta \quad (18)$$

Fig. R3a. b and c illustrate the theoretical result, simulation and experimental measurements of the membrane's deformation in z direction upon the illumination of the pulsed laser at a power density of 1 MW/cm². demonstrate the overall consistency of the results obtained from the three methods. This verification confirms the validity of our theoretical deductions. As illustrated in Fig. R3, the pulsed laser excites localized Lamb waves within the thin film, which gradually propagate outwards over time. These localized Lamb waves have the capability to displace particles from the thin film, allowing them to arrange themselves into arbitrary patterns.

Fig. R3. (appears as S. Fig. 3 of the Supplementary Information) a) Theoretical calculation of elastic wave propagation depicted in space-time diagrams. b) Numerical simulation of elastic wave propagation depicted in space-time diagrams. c) Experimental measurements of elastic wave propagation depicted in space-time diagrams.

During the derivation of the formulas, it is evident that the intensity of the laser has a linear impact on both the temperature increase of the metal membrane and the intensity of vibration in the metal membrane.

Fig. R4. (appears as S. Fig. 4 of the Supplementary Information) a) The maximum temperature of the multilayer membrane varies with changes in laser power density. b) The maximum displacement of the multilayer membrane varies with changes in laser power density c) Theoretical displacement of the multilayer membrane at $x = 0.5$ mm exhibits variations with changes in both time and laser power density. d) Theoretical acceleration of the multilayer membrane at $x = 0.5$ mm exhibits variations with changes in both time and laser power density.

Fig. R4a illustrates the relationship between the light intensity and the calculated maximum temperature increase in the membrane, comparing the theoretical and simulated results. Similarly, Fig. R4b displays the relationship between the light intensity and the calculated maximum vibration displacement in the membrane, again comparing the theoretical and simulated results. These findings align with our initial expectations. It is important to note that despite the temperature within the membrane exceeding 100 °C, it rapidly dissipates to room temperature and does not reach a boiling point that would affect the water layer.

In Fig. R4c, we presented the variations of displacement with respect to time and laser power density at the coordinate $x = 0.5$ mm. Additionally, in Fig. R4d, we displayed the corresponding variations of acceleration. It is observed that the vibration field at the point located at coordinate $x = 0.5$ mm on the membrane demonstrates a linear correlation with the incident laser power density at all time instances.

The silica particles positioned on the multilayer membrane undergo distinct force states as a result of the varying motion of the membrane. This phenomenon is depicted in Fig. R5, which presents a schematic diagram illustrating the concept.

Fig. R5. (appears as S. Fig. 8 of the Supplementary Information) **The force state of a silica particle on the multilayer membrane.** The positive and negative signs in red represent the direction of acceleration of the multilayer membrane, with upward indicating the positive direction. a) Net force acting on the particles is zero in stationary state. b) Net force acting on the particles is directed downward when multilayer membrane is accelerating downward. c-d) Net force acting on the particles is directed upward when acceleration of multilayer membrane is upward. e) Net force acting on the particles is directed downward, and the distance between the particles and the membrane increases during this stage. f) Net force is solely provided by gravity and contact pressure as the distance continues to increase.

During the stage illustrated in Fig. R5a-d, where the particles vibrate in sync with the multilayer membrane, the net force acting on the particles can be determined by considering the acceleration of the multilayer membrane and the mass of the particles. In Fig. R6, the red line represents the net force acting on the particle located at the coordinate $x = 0.5$ mm. The corresponding laser power density in this case is $2.0 \text{ MW}/\text{cm}^2$. The detachment point occurs between Fig. R5d and Fig. R5e, when the upward velocity of the membrane reaches its maximum. Subsequently, both the membrane and particle velocities decrease. However, if the membrane decelerates faster than the particles, the distance between the membrane and particles will increase.

In Fig. R5e-f, it can be observed that as the particle starts to detach from the multilayer membrane, the pressure exerted on the particles by the membrane gradually decreases, while the acceleration of the particles remains negative throughout this process. To calculate the net force acting on the particles, it is necessary to determine the distance between the particles and the membrane. In Supplementary Note 4, we utilized numerical integration to calculate the motion of the particles during the detachment

process, enabling the determination of the net force on the particles. The calculated result after detachment is represented by the blue line in Fig. R6.

Fig. R6. (appears as S. Fig. 9 of the Supplementary Information) **Net force acting on particles at the coordinate $x = 0.5$ mm.**

References

- 1 Chen X., Chen Y. T., Yan M., and Qiu M. Nanosecond Photothermal Effects in Plasmonic Nanostructures, *ACS Nano* **6** (3), 2550-2557 (2012).
- 2 Auld, B. A., and R. E. Green. Acoustic fields and waves in solids. *Physics Today* **27**(10), 63-64 (1974).
- 3 Rossignol C., Rampnoux J. M., Perton M., Audoin B., and Dilhaire S. Generation and Detection of Shear Acoustic Waves in Metal Submicrometric Films with Ultrashort Laser Pulses. *Phys. Rev. Lett.* **94**, 166106 (2005).
- 4 Nagy P. B. and Nayfeh A. H. Viscosity-induced attenuation of longitudinal guided waves in fluid-loaded rods, *The Journal of the Acoustical Society of America*, **100** (3), 1501-1508 (1996).

Summary of the modifications:

- (1) On Page 3, marked by deep red.

~~“The influence of laser power density on the temperature and deformation of the multilayer film is linear. Increasing the power density results in larger displacements of the multilayer membrane, which in turn affects the forces acting on the particles and their motion state. The antisymmetric mode of Lamb wave can generate a larger amplitude compared to symmetric mode, making it useful for propelling an object.”, “The configuration of localized Lamb waves closely resembles the optical intensity distributions on the membrane.”.~~

(2) On Page 5, marked by deep red.

~~“Through theoretical derivation, we can obtain the space-time diagrams of membrane’s deformation similar to Fig. 1b, c. The results are shown in S. Fig. 3.”.~~

(3) On Page 7, marked by deep red.

~~“In Supplementary Note 4, we analyzed the different force states of particles on the multilayer film and the characteristics of the detachment critical point. Then, using numerical integration methods, we obtained the variation of the force exerted on the particles with time and their motion state. As the membrane undulates, t Particles that meet the separation critical condition are lifted, then scatter, and fall onto the surrounding areas.”.~~

(4) On Page 14, marked by deep red.

~~“**Theoretical derivation of photoacoustic.** The photoacoustic effect involves the conversion of light energy into heat energy, and heat energy into mechanical energy. In the derivation of the opto-thermal process, we first calculated the temporal variation of the surface temperature of the multilayer membrane. Using this as a boundary condition, we then computed the diffusion of heat and obtained the spatiotemporal distribution of the temperature within the multilayer membrane. By calculating the strain caused by temperature changes, we obtained the initial displacement. The multilayer film under laser stripe illumination can be approximated as an infinitely long beam. We then applied the initial displacement to the vibration equation of the infinitely long beam to obtain the space-time diagrams of membrane’s deformation. From the derivation We find that the laser power density linearly effects the temperature and vibration displacement of the multilayer membrane (detailed in Supplementary Note 1).”.~~

(5) We have added Figs. R1-6 in the revised Supplementary Information, on Page 1-17, S. Fig. 1-4, 8, 9, to clarify the impact of particle types on pattern formation.

Reviewer #3 (Remarks to the Author):

The author has successfully responded to the reviewers' comments. I believe that the manuscript's quality has improved sufficiently for me to propose it for publication in this journal.

We greatly appreciate the high evaluation of our manuscript by the reviewer.

Reviewers' Comments:

Reviewer #2:

Remarks to the Author:

The author has successfully responded to the reviewers' comments. I believe that the quality of the manuscript has now improved sufficiently for me to propose it for publication in this journal.